# Quantification of *Pseudomonas aeruginosa* in multispecies biofilms using PMA-qPCR

Sarah Tavernier and Tom Coenye

Laboratory of Pharmaceutical Microbiology, Ghent University, Ghent, Belgium

## ABSTRACT

Multispecies biofilms are an important healthcare problem and may lead to persistent infections. These infections are difficult to treat, as cells in a biofilm are highly resistant to antimicrobial agents. While increasingly being recognized as important, the properties of multispecies biofilms remain poorly studied. In order to do so, the quantification of the individual species is needed. The current cultivation-based approaches can lead to an underestimation of the actual cell number and are time-consuming. In the present study we set up a culture-independent approach based on propidium monoazide qPCR (PMA-qPCR) to quantify *Pseudomonas aeruginosa* in a multispecies biofilm. As a proof of concept, we explored the influence of the combined presence of *Staphylococcus aureus*, *Streptococcus anginosus* and *Burkholderia cenocepacia* on the antimicrobial susceptibility of *P. aeruginosa* using this PMA-qPCR approach.

## INTRODUCTION

Specific quantification of the different members in a multispecies biofilm is a challenging task. Cultivation-based approaches are time-consuming and can lead to an underestimation of cell numbers due to the presence of viable but nonculturable bacteria (VBNC). VBNC bacteria will not grow on routinely-used microbiological media, but are nevertheless still viable and potentially virulent (*Li et al., 2014*). A promising alternative for cultivation-based methods is quantification based on qPCR. However, a major drawback of qPCR-based quantification is the overestimation of cell numbers due to the presence of extracellular DNA and DNA originating from dead cells, and adjustments are required to differentiate between viable and dead bacteria. Treatment of bacterial samples with propidium monoazide (PMA) prior to DNA extraction has been proposed as an effective method to avoid the detection of extracellular DNA and DNA from dead cells (*Alvarez et al., 2013*; *Kruger et al., 2014*; *Yasunaga et al., 2013*). PMA only enters membrane-compromised cells, and once inside the cell, it intercalates into DNA between the bases (one PMA molecule per 4 to 5 base pairs DNA, with little or no sequence preference). Besides intercalating into DNA of membrane-compromised cells, PMA can also intercalate into extracellular DNA (*Nocker, Sossa & Camper, 2007*; *Waring, 1965*). After exposure to strong visible light, the photoreactive azido group of PMA is converted to a

Corresponding author
Tom Coenye,
Tom.Coenye@Ugent.be

reactive nitrene radical. This nitrene radical forms a stable covalent nitrogen-carbon bond with the DNA, resulting in permanent DNA modification. The modified DNA is then lost together with cells debris during genomic DNA extraction and will not be amplified during the qPCR reaction (*Nocker, Cheung & Camper, 2006*; *Nocker et al., 2009*). Excess PMA is inactivated by reaction with water molecules in solution, prior to DNA extraction, and thus will not affect the DNA from viable cells after cell lysis (*Nocker, Cheung & Camper, 2006*). Nevertheless, the use of PMA has some limitations. The discrimination between viable and dead cells is only based on membrane integrity, and the effect of antimicrobial therapies that do not target the cell membrane can thus not be monitored using PMA (*Nocker & Camper, 2009*). Secondly, viable cells with a slightly damaged cell membrane will not be quantified (*Strauber & Muller, 2010*) and the presence of a high number of dead cells ($>10^4$ cells/ml) can affect the quantification of viable cells (*Fittipaldi, Nocker & Codony, 2012*). Finally, the presence of other compounds in the sample, e.g., environmental compounds that can bind to PMA, can subsequently prevent PMA to bind to DNA (*Taylor, Bentham & Ross, 2014*).

In order to determine whether there are differences in antimicrobial susceptibility of cells grown in mono-or multispecies biofilms (*Dalton et al., 2011*; *Lopes et al., 2012*), accurate quantification of the various members of these biofilms is required. In the present study, we evaluated the use of PMA-qPCR to quantify *Pseudomonas aeruginosa* in mono- and multispecies biofilms following exposure to various antibiotics, used to treat respiratory infections in cystic fibrosis (CF).

## MATERIALS AND METHODS

### Bacterial strains

*P. aeruginosa* ATCC9027, *Staphylococcus aureus* LMG10147, *Burkholderia cenocepacia* LMG16656, and *Streptococcus anginosus* LMG14502 were cultured overnight at 37 °C in Brain Heart Infusion broth (BHI) (Oxoid, Basingstoke, UK).

### Antibiotic solutions

The difference in susceptibility of *P. aeruginosa* in a mono- and multispecies biofilm towards colistin (Sigma-Aldrich, Diegem, Belgium), tobramycin (TCI Europe, Zwijndrecht, Belgium), and levofloxacin (Sigma-Aldrich, Diegem, Belgium) was determined. The concentration used for colistin and tobramycin was 200 µg/ml, for levofloxacin 100 µg/ml. These concentrations were based on the levels achievable in CF sputum by inhalation therapy (*Wu et al., 2013*). All antibiotic solutions were prepared in physiological saline (0.9% [w/v] NaCl) (PS) (*Keltner et al., 2014*). Minimal inhibitory concentrations (*Nusbaum et al., 2012*) of colistin, tobramycin and levofloxacin were determined in duplicate according to the EUCAST broth microdilution protocol using flat-bottom 96-well microtiter plates (TPP, Trasadingen, Switzerland) as previously described (*Peeters, Nelis & Coenye, 2008*).

## Quantification of *P. aeruginosa* cells in monospecies and multi-species biofilms

### Formation of P. aeruginosa monospecies and multispecies biofilms

For formation of mono- and multispecies biofilms, round-bottomed 96-well microtiter plates (TPP) were used. Inoculum suspensions containing approximately $10^6$ CFU/ml of *P. aeruginosa* alone or $10^6$ CFU/ml of *P. aeruginosa* in combination with $10^6$ CFU/ml of *S. aureus*, $10^7$ CFU/ml of *B. cenocepacia* and $10^7$ CFU/ml of *S. anginosus*, were made in BHI. The inoculum cell numbers were based on preliminary optimization experiments, and led to biofilms with the highest cell numbers (data not shown). BHI was supplemented with 5% (w/v) bovine serum albumin (BSA) (*Kart et al., 2014*), 0.5% (w/v) mucine type II, and 0.3% (w/v) agar. Mucine and agar were added to mimic the composition of CF sputum and to increase the medium viscosity, respectively. Sterile medium served as blank and was included on each plate. After 4 h of adhesion at 37 °C, wells were rinsed with 100 µl PS to remove non-adhered cells. A 100 µl amount of fresh medium was added to the wells and the plates were incubated for an additional 20 h. After 20 h, the supernatant was again removed, each well was rinsed using 100 µl PS and 100 µl of the antibiotic solution (colistin, tobramycin or levofloxacin) was added to the mature biofilms. To the wells of the control biofilm plate, 100 µl PS was added. The plates were then again incubated at 37°C for 24 h. For each test condition, 72 technical replicates were included. All experiments were performed on three different occasions. Confocal imaging was performed as described in *Udine et al. (2013)*. The control plate cell numbers of *P. aeruginosa*, *S. aureus* and *B. cenocepacia*, respectively, determined on cetrimide agar, mannitol salt agar and tryptic soy agar supplemented with tobramycin (4 mg/ml) and nitrofurantoin (25 mg/ml), increased after 24 h, respectively with 1.30, 0.67 and 0.95 log cfu/biofilm, indicating that these species are actually growing in the multispecies biofilm. The control cell number of *S. anginosus,* determined on Mc Kay agar (*Sibley et al., 2010*) did not change, indicating that this species was continuously present in the multispecies biofilm. Viable but nonculturable cells of any species might not be detectable by plate counts, which might lead to differences between plate counts and PMA-qPCR.

### Propidium monoazide cross-linking

After 24 h of experimental administration of antibiotics to the biofilm, the antibiotic solution in the test plate and the PS in the control plate were removed. The wells were rinsed with 100 µl PS. Next, biofilms were detached by vortexing (900 rpm) and sonication (both 5 min), followed by collection of the content of the wells in a sterile tube. The vortexing and sonication step was repeated after the addition of 100 µl PS to each well. The sterile tube was centrifuged (5 min at 3000 × g), and the pellet was resuspended in 1.5 ml of PS. For each treatment, 2 wells of a 24-well plate were filled with 600 µl of the cell suspension. 1.5 µl of a 20 mM PMA solution in dH$_2$O (Biotium, Inc., California, USA) was added to the first well (final concentration of 50 µM). To the second well, 1.5 µl of MilliQ water (MQ water) (Millipore, Billerica, Massachusetts, USA) was added. The plates were vortexed (5 min, 300 rpm, room temperature) in the dark and exposed to light for 10 min,

using a LED-lamp (Dark Reader transilluminator, Clare Chemical Research, Dolores, Colorado, USA) (output wavelength 465–475 nm) (*Deschaght et al., 2013*).

*Effect of PMA on P. aeruginosa cell viability.* To analyze the effect of 50 μM PMA on cell viability, overnight grown planktonic *P. aeruginosa* cells ($OD_{600} = 1.0$) were used. To 2 mL of this culture, 5 μl of a 20 mM PMA solution in $dH_2O$ was added (final concentration of 50 μM). As a control, 5 μl of MQ water was added instead of the PMA solution. The plates were incubated in the dark (5 min, 300 rpm) and exposed to light for 10 min. Next, the cell numbers of control and test group were determined via the plate count method (on tryptic soy agar) and by solid-phase cytometry (SPC) (ChemScan RDI; AES-Chemunex, Ivry-sur Seine, France), as described previously (*Vanhee et al., 2010*).

### Extraction of genomic DNA

After incubation with PMA, 500 μl of cell suspension from each well was transferred to a sterile Eppendorf tube. The samples were centrifuged (10 min, 13.000 rpm) and DNA from Gram-negative organisms was extracted as described previously (*Pitcher, Saunders & Owen, 1989*). Briefly, the pellets were washed with 500 μl RS-buffer (0.15M NaCl [Sigma-Aldrich, Diegem, Belgium], 0.01 M EDTA [VWR, Leuven, Belgium], pH 8.0) and resuspended in TE-buffer (1 mM EDTA, 10 mM Tris–HCl [Sigma-Aldrich, Diegem, Belgium]). 500 μl GES-buffer (60% [w/v] guanidium thiocyanate [Sigma-Aldrich, Diegem, Belgium], 0.5 M EDTA, pH 8.0, 1% [w/v] sarkosyl [Sigma-Aldrich, Diegem, Belgium]) was added and the samples were placed on ice for 10 min. After the addition of 250 μl cold ammonium acetate (7.5 M [VWR, Leuven, Belgium]), the samples were placed back on ice for 10 min. Subsequently, 500 μl cold chloroform/isoamylalcohol (24:1) (Roth, Karlsruhe, Germany) was added. Samples were mixed thoroughly and centrifuged for 20 min at 13.000 rpm. Supernatant was then collected in a new tube and 0.54 volumes cold isopropanol (Sigma-Aldrich, Diegem, Belgium) were added to precipitate the DNA. Samples were then centrifuged (10 min, 13.000 rpm), and supernatant was removed. A 150 μl amount of ethanol (Sigma-Aldrich, Diegem, Belgium) (70% [v/v]) was added and samples were centrifuged for 1 min. This step was repeated. The DNA pellet was air-dried, 30 μl TE-buffer was added, the samples were placed at 4° C for 24 h and were then treated with RNase.

Following electrophoresis on 1% agarose gels, genomic DNA was visualized with GelRed (GelRed nucleic acid, Biotium, Hayward, California, USA), and the genomic DNA concentration was measured with Quantifluor dsDNA kit (Promega, Madison, Wisconsin, USA).

### qPCR

Real-time PCR (CFX96 Real Time System; Bio-Rad, Hercules, CA, USA) was carried out with the PerfeCTa SYBR Green FastMix (Quanta Biosciences). Species-specific primer sequences for *gyrB* of *P. aeruginosa* were designed using primer-BLAST (http://www.ncbi.nlm.nih.gov/tools/primer-blast/) using *P. aeruginosa* and *B. cenocepacia* sequences obtained from GenBank. The forward primer and the reverse primer were

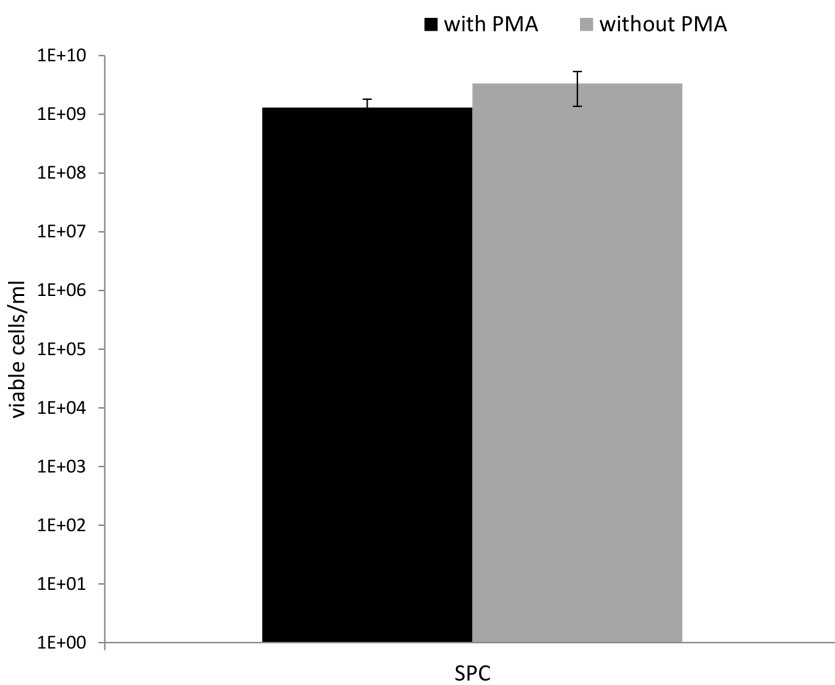

**Figure 1 Number of viable cells (determined using SPC) in PMA-treated (50 μM) and untreated samples.** Treatment with PMA (50 μM) did not affect the number of viable cells (Mann-Whitney test, $p > 0.05$).

5′-GGTGTTCGAGGTGGTGGATA-3′ and 5′-TGGTGATGCTGATTTCGCTG-3′, respectively. The specificity of the primers was evaluated by melting curve analysis.

To generate a standard curve, DNA extracted from serially-diluted and PMA-treated planktonic *P. aeruginosa* cultures was used for qPCR. The $C_q$-values obtained were plotted against the number of viable cells determined by SPC. The serial dilutions were prepared from a *P. aeruginosa* overnight suspension (OD 0.1). Cells were diluted from $10^9$ CFU/ml to $10^4$ CFU/ml in PS. Six independent biological repeats were included.

### Effect of PMA on $C_q$-values of defined ratios of viable and dead *P. aeruginosa* cells

Planktonic *P. aeruginosa* cells (OD 1.0) were killed by heating for 15 min at 95°C. Complete killing was confirmed by SPC. Mixtures of viable and dead cells were prepared, in which viable cells represented 0%, 0.1%, 1%, 10%, 50%, 75% and 100% of the total population. Four wells of a 24-well plate were filled with 600 μl of each mixture. PMA was added to 2 wells and MQ water was added to the other 2 wells (PMA-negative control). Cells were then treated as described above (2.3.2.) and $C_q$-values were determined via qPCR. Six independent biological repeats were carried out.

### Statistical data analysis

Statistical data analysis was performed using SPSS software, version 22 (SPSS, Chicago, Illinois, USA). The normal distribution of the data was verified using the Shapiro–Wilk test. Non-normally distributed data were analyzed using a Mann–Whitney test. Normally

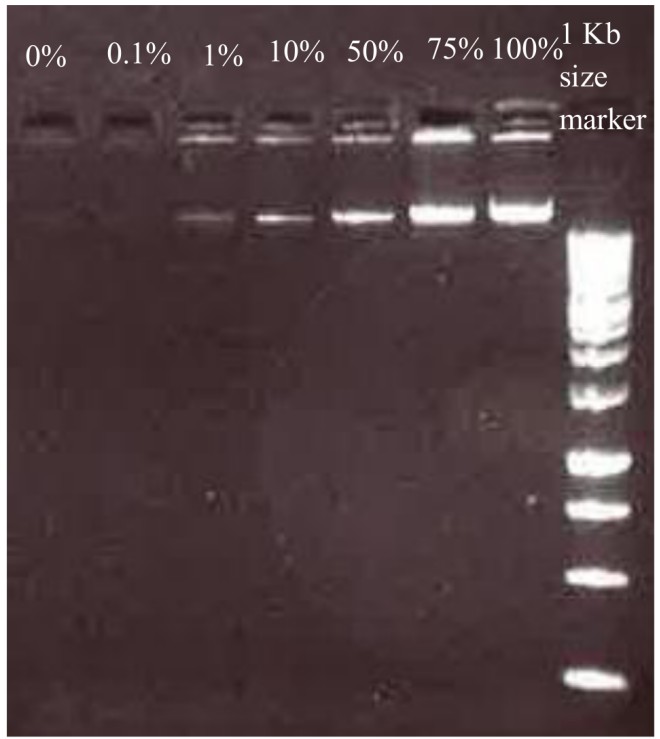

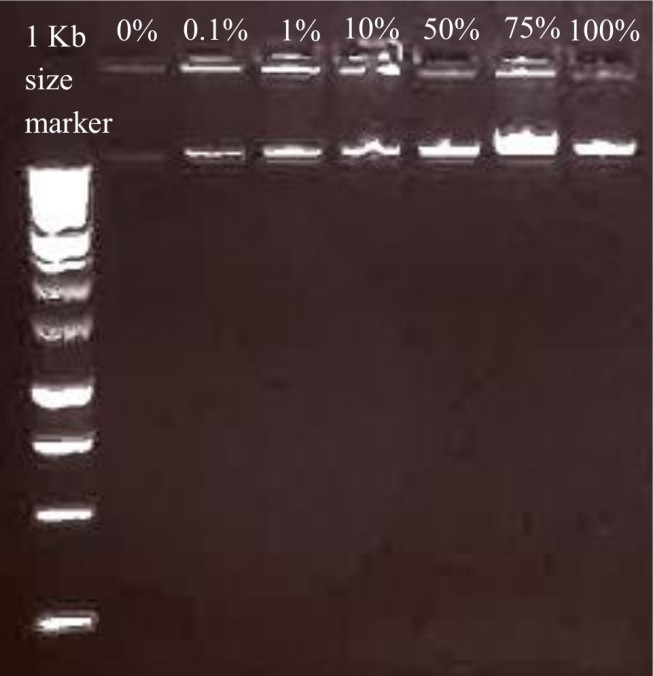

**Figure 2 Agarose gel with DNA from PMA-treated and untreated mixtures of living and dead cells.** Genomic DNA extracted from PMA-treated mixtures (A) and PMA-untreated mixtures (B), containing an increasing fraction of viable *P. aeruginosa* cells.

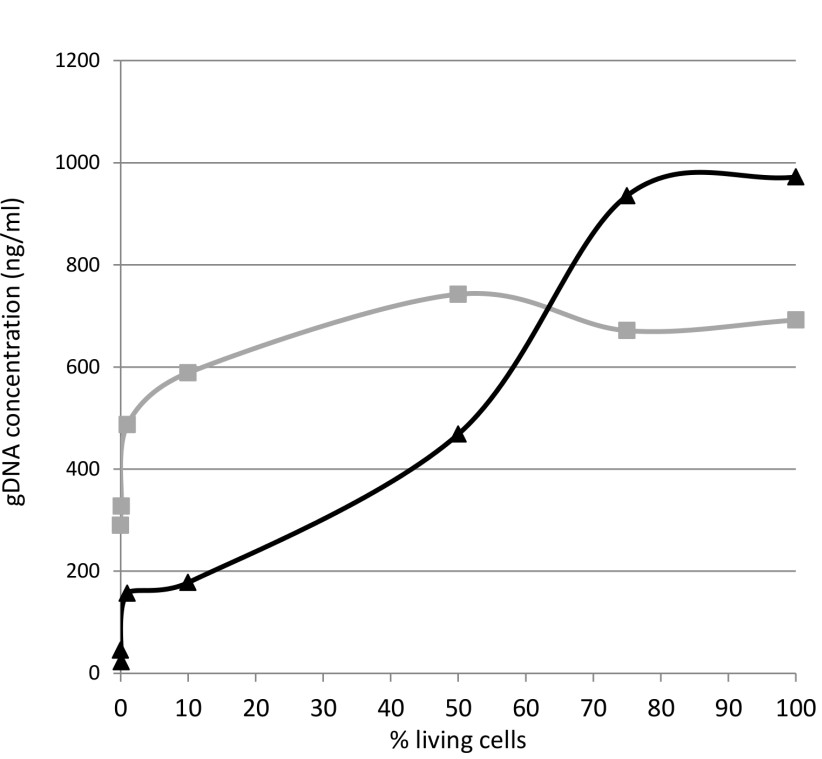

**Figure 3** **Correlation between the gDNA concentration (ng/ml) and the percentage of living cells.** Data were obtained with the same samples used in Figs. 2A and 2B.

distributed data were analyzed using an independent sample $t$-test. Differences with a $p$-value $<0.05$ were considered as significant.

## RESULTS AND DISCUSSION

### Optimization of the PMA-qPCR

Treatment with PMA (50 μM) did not affect the number of viable cells as determined via SPC (Fig. 1), so it can be concluded that PMA itself has no inhibitory effect on *P. aeruginosa*. Therefore, all experiments were conducted with a PMA concentration of 50 μM. Increasing the fraction of viable cells in the mixture led to an increase in the genomic DNA yield after PMA treatment (Figs. 2A, 2B and 3). As shown in Fig. 2A, the DNA concentration increases with an increasing percentage of living cells, after PMA treatment. Without PMA treatment, the DNA concentration between all mixtures was more similar (Fig. 2B). This indicates that the DNA of the heat-killed cells is still present in PMA-untreated mixtures. The correlation between the DNA concentration and the percentage of living cells is shown in Fig. 3. The same trend can be seen as in (Figs. 2A, 2B and 3): the DNA concentration in the PMA-treated mixtures is increasing with an increasing number of living cells, while the DNA concentration in the PMA-untreated mixtures is higher for a lower percentage of living cells and more quickly reached a plateau phase. This indicates that the DNA concentration (and subsequently the viable cell

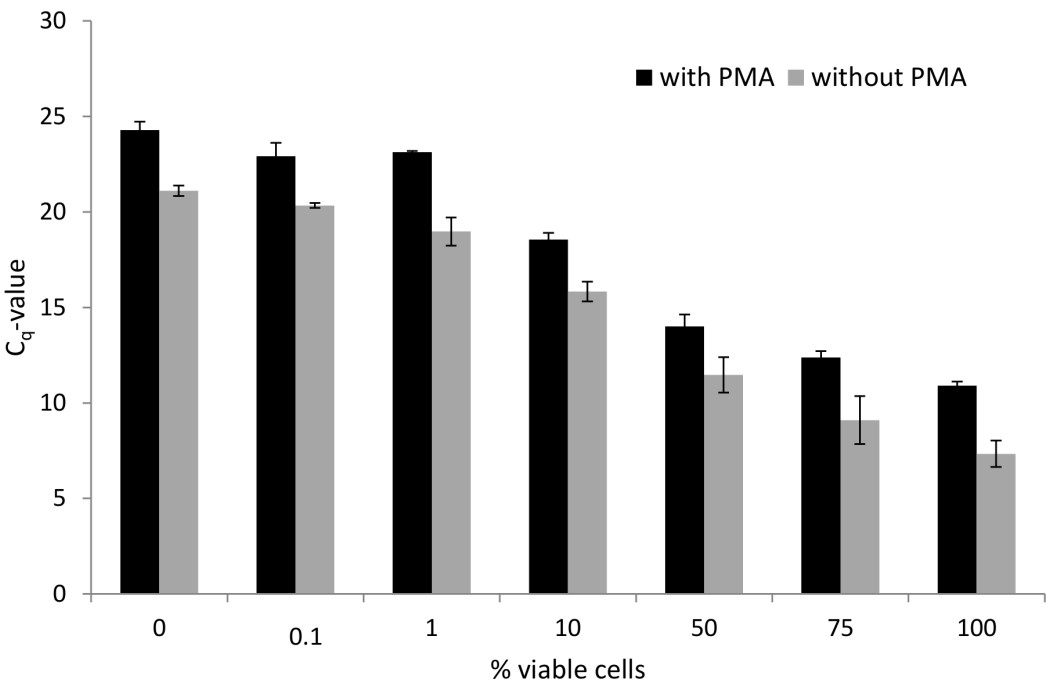

**Figure 4 Effect of PMA treatment on Cq-values obtained following qPCR using DNA extracted from mixtures of viable and heat-killed *P. aeruginosa* cells.** The addition of PMA leads to a higher Cq-value. This indicates that the amplification of DNA of heat-killed cells was inhibited by PMA. Error bars represent the standard error mean ($n = 6$). (*: $p < 0.05$, Mann-Whitney test).

number) is less overestimated in PMA-treated samples. Increasing the fraction of viable cells in the mixture also led to a significant decrease of the $C_q$-value determined via qPCR (Fig. 4). The $C_q$-values obtained with PMA-treated mixtures were significantly higher than the $C_q$-values of corresponding PMA-untreated mixtures ($p < 0.05$). The higher $C_q$-value after PMA treatment indicates that the qPCR amplification of DNA of heat-killed cells is efficiently inhibited by the addition of PMA. This is confirmed by the decrease in $C_q$-value after increasing the fraction of viable cells and was also described by *Alvarez et al. (2013)*. Without PMA treatment, there is also a decrease in $C_q$-value after increasing the fraction of viable cells: actively proliferating bacterial cells have more DNA than inactive or dead cells, which lead to a lower $C_q$-value when the percentage of living cells is increased, even without the addition of PMA. Moreover, the dead cells in the samples were obtained using heat inactivation (95 °C, 15 min). This heat treatment can have an influence on the degradation of the DNA, and may subsequently lead to a lower amount of intact DNA and thus a higher $C_q$-value when the percentage of inactivated cells increases over live cells (*Takahashi et al., 2004*).

When plotting the log of the number of viable cells versus $C_q$-values obtained, a linear relationship was observed between both parameters ($R^2 = 0.9685$) (Fig. 5). The linear range of this relationship is between $10^5$ and $10^9$ cells, indicating that the method used is limited to treatments that result in a number of surviving cells higher than $10^5$. A viable cell number of $10^5$ corresponds to a $C_q$-value of approximately 30. *Nocker et al. (2009)*

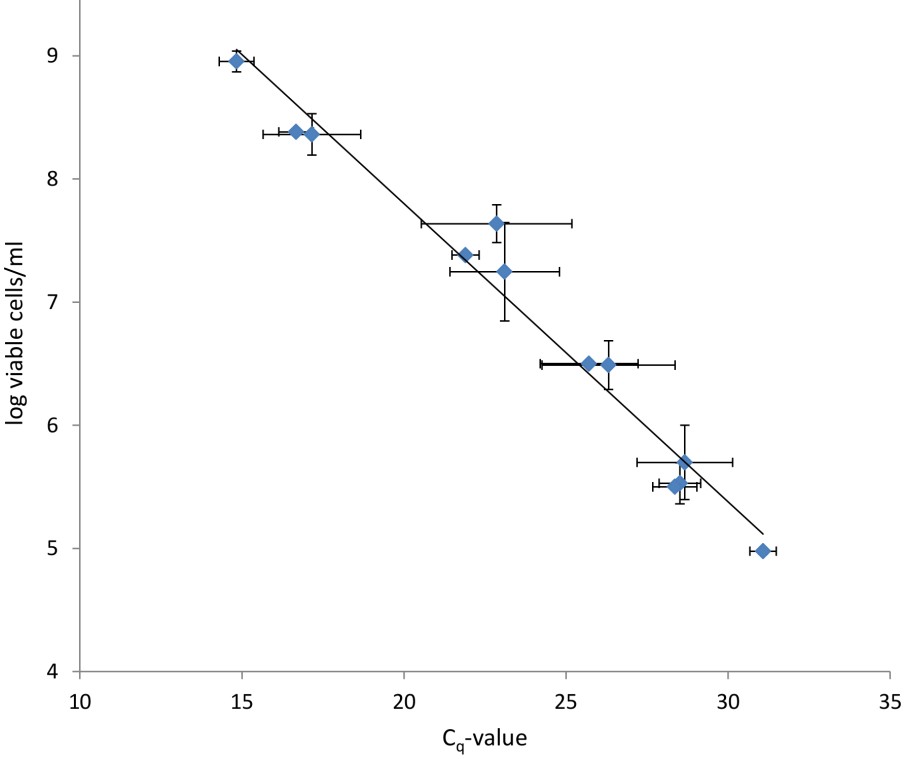

**Figure 5 Correlation between log viable P. aeruginosa cells/ml determined via SPC and Cq-values determined via PMA-qPCR.** The equation for the linear trendline is $y = -0.2421x + 12.642$ with $R^2 = 0.9685$. Using this equation, the log viable cells/ml can be calculated from the Cq values obtained with PMA-qPCR. Since one biofilm represents a volume of 200 μl, the log viable cells/biofilm can be calculated by dividing the log viable cells/ml by five. Error bars represent standard deviations ($n = 6$).

described that signals from killed cells could not be suppressed completely by PMA at very low ratios of live/killed cells, with corresponding $C_q$-values of 30 or higher. This could be due to the sensitivity of exponential amplification, and could be a possible explanation for the lower limit of the linear range.

In native material of multispecies biofilms, for instance from patients' material, divergent results might occur, and further testing of spiked samples may be useful to clarify this point of uncertainty also raised by *Taylor, Bentham & Ross (2014)*.

## Susceptibility of planktonic and sessile *P. aeruginosa* cells to colistin, levofloxacin, and tobramycin

The MIC of tobramycin for *P. aeruginosa* ATCC 9027 planktonic cells was 0.5 μg/ml, the MIC of colistin was 2 μg/ml and the MIC of levofloxacin was 1 μg/ml. These concentrations are below the breakpoint for *P. aeruginosa* (National Committee for Clinical and Laboratory Standards, 2007), indicating that *P. aeruginosa* is sensitive to the antibiotics used.

A confocal image of the mature multispecies biofilm is shown in Fig. 6. Both rod-shaped bacteria (presumed Gram-negative bacteria—*P. aeruginosa* and *B. cenocepacia*)—and coc-

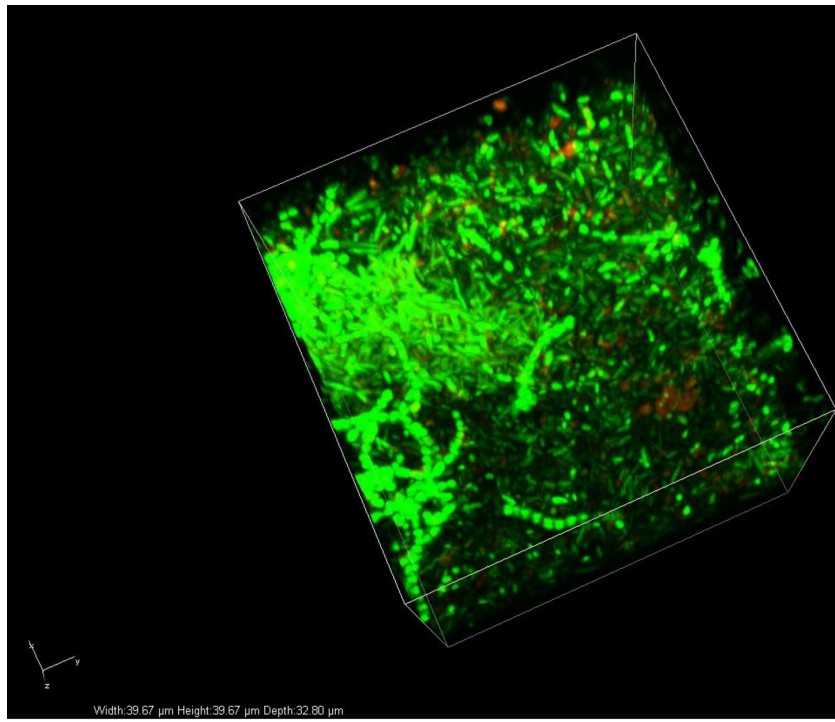

Width:39.67 µm Height:39.67 µm Depth:32.80 µm

**Figure 6 Microscopy image of a multispecies biofilm.** A confocal image of a mature multispecies biofilm (Live/Dead staining). Rod-shaped and coccal bacteria are clearly visible in the biofilm image.

cal bacteria (presumed to be the Gram-positive cocci—*S. aureus* and *S. anginosus*)—were visible.

The susceptibility of sessile *P. aeruginosa* cells in mono- and multispecies biofilms to antibiotics was determined with PMA-qPCR. The reduction of the number of viable cells was calculated by using the equation for the linear trendline describing the relationship between the log viable *P. aeruginosa* cells/ml and the $C_q$-value obtained with PMA-qPCR. Colistin treatment (200 µg/ml, 24 h) led to a significant decrease ($p < 0.05$) in the number of viable *Pseudomonas aeruginosa* cells, both in mono- and multispecies biofilms (Fig. 7 and Table 1). In multispecies biofilms, this average reduction was 1.26 log. Using the plate count method, an average reduction of 1 log was observed in *P. aeruginosa* monospecies biofilms. However, based on PMA-qPCR, more viable cells were present, suggesting that the use of the plate count method leads to an underestimation of the surviving cell numbers. The results also show that *P. aeruginosa* is significantly more sensitive to colistin in a multispecies biofilm with *S. aureus*, *S. anginosus* and *B. cenocepacia* than in a monospecies biofilm ($p < 0.05$). After treatment with levofloxacin (100 µg/ml, 24 h), there was also a significant reduction in the number of viable *P. aeruginosa* cells, both in mono- and multispecies biofilms ($p < 0.05$) (Fig. 7). Based on the equation for the linear trend line describing the relationship between the log viable *P. aeruginosa* cells/ml and the $C_q$-value obtained with PMA-qPCR, a 1.57 log reduction was observed for *P. aeruginosa* in monospecies biofilms, while in multispecies biofilms, this average reduction was only

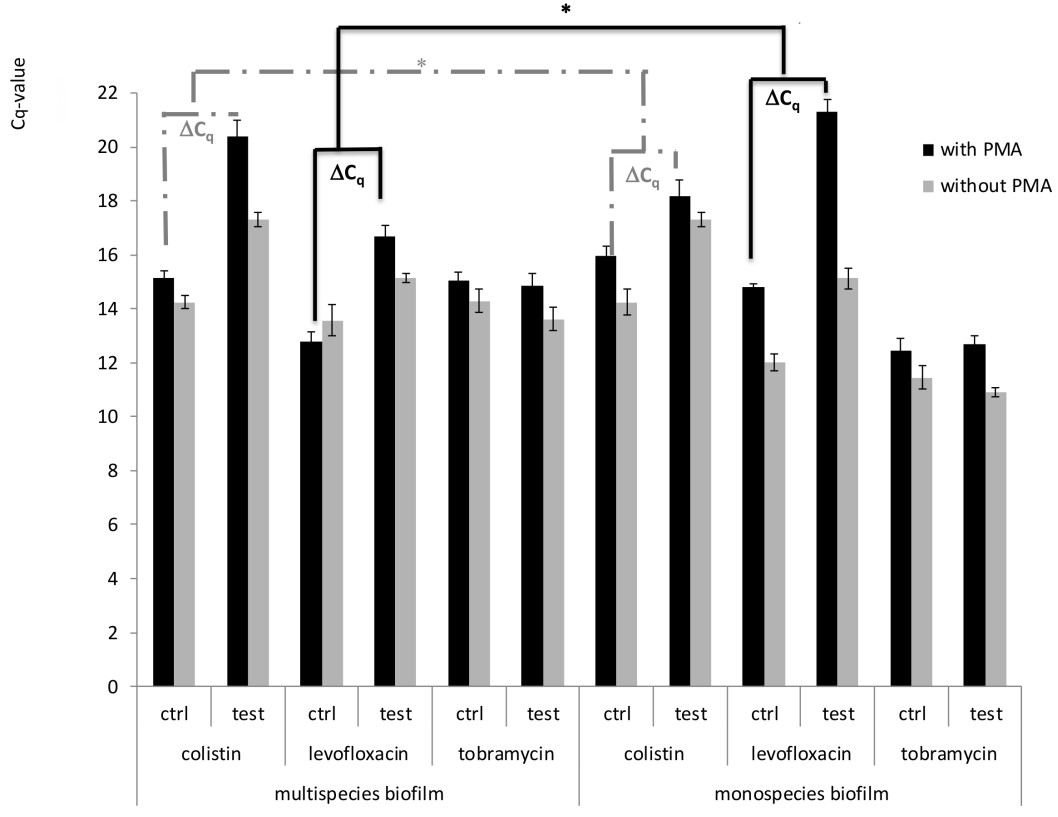

**Figure 7 Cq-values obtained with PMA-qPCR of DNA samples recovered from mono- or multi-species biofilms, after treatment with colistin (200 μg/ml), levofloxacin (100 μg/ml) and tobramycin (200 μg/ml) for 24 h.** Error bars represent standard error mean values ($n = 3 \times 2$). (*: $p < 0.05$)

0.94 log ($p < 0.05$) (Table 1). These results indicate that *P. aeruginosa* is less susceptible to levofloxacin in a multispecies biofilm.

For tobramycin (200 μg/ml, 24 h), there was no significant increase in $C_q$-values after treatment ($p > 0.05$) (Fig. 7 and Table 1). Nevertheless, experiments using the plate count method showed an average reduction of *P. aeruginosa* in monospecies biofilms of 2.35 log after treatment with tobramycin (Fig. 8). A likely explanation is that tobramycin causes little or no loss of membrane integrity (*Kim et al., 2008*; *Tack & Sabath, 1985*). Bacterial cells can be killed by tobramycin, but their DNA can still be amplified in the qPCR reaction, as PMA cannot bind to the genomic DNA of intact cells. DNA of dead cells is then extracted together with the DNA of living cells in the DNA extraction procedure and amplified during qPCR, resulting in a lower $C_q$-value and an overestimation of the number of viable cells.

## CONCLUSIONS

The present study shows that PMA-qPCR is a useful alternative for the plate count method to quantify *P. aeruginosa* in mono- and multispecies biofilms, after treatment with a membrane-compromising agent. This method can thus be used to avoid underestimating

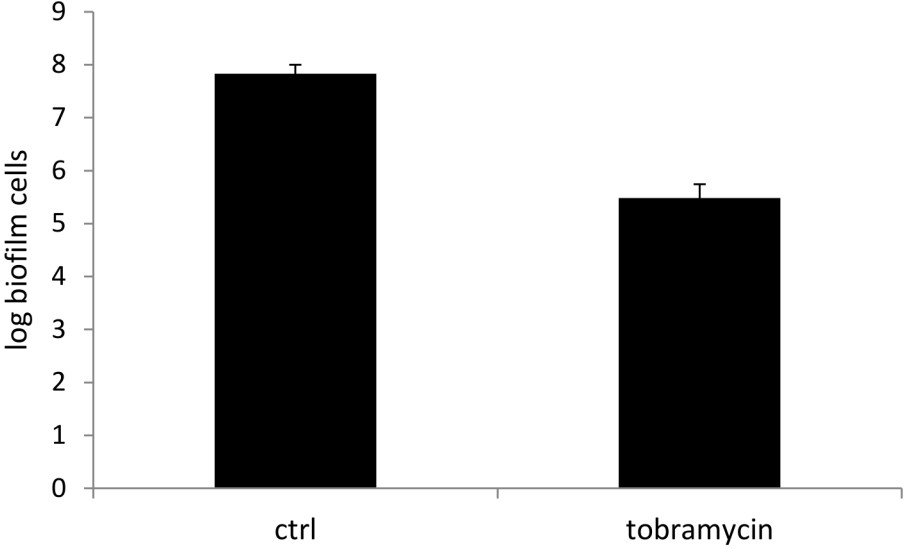

**Figure 8 Number of cells in treated and untreated biofilms.** Log P. aeruginosa biofilm cells in a control biofilm and in a test biofilm, after treatment with tobramycin (200 µg/ml) for 24 h, determined with the plate count method. Errors bars represent standard deviation values ($n = 3 \times 2$) ($p < 0.05$).

**Table 1 Comparison of $C_q$ values and calculated number of CFU.** $C_q$-values (±SEM) obtained with PMA-qPCR and calculated log values of viable *P. aeruginosa* biofilm cells (using the equation for the linear trendline), after treatment with colistin (200 µg/ml), levofloxacin (100 µg/ml) or tobramycin (200 µg/ml) for 24 h and incubation with PMA. The difference in the calculated number of viable cells after treatment is significantly different between mono- and multispecies biofilms ($p < 0.05$).

| Biofilm type | Treatment | $C_q$-value (±SEM) | Calculated log viable cells/biofilm | Δlog |
|---|---|---|---|---|
| Multispecies | – | 15.13 ± 0.28 | 8.27 | 1.26 |
| | Colistin | | 7.01 | |
| | | 20.39 ± 0.64 | | |
| Monospecies | – | 15.98 ± 0.35 | 8.07 | 0.53 |
| | Colistin | | 7.54 | |
| | | 18.18 ± 0.23 | | |
| Multispecies | – | 12.79 ± 0.38 | 8.85 | 0.95 |
| | Levofloxacin | | 7.90 | |
| | | 16.71 ± 0.38 | | |
| Monospecies | – | 14.80 ± 0.15 | 8.36 | 1.57 |
| | Levofloxacin | | 6.79 | |
| | | 21.29 ± 0.46 | | |
| Multispecies | – | 15.04 ± 0.44 | 8.30 | −0.05 |
| | Tobramycin | | 8.35 | |
| | | 14.84 ± 0.43 | | |
| Monospecies | | 14.80 ± 0.15 | 8.36 | −0.02 |
| | Tobramycin | | 8.38 | |
| | | 14.70 ± 0.18 | | |

the cell number due to the presence of VBNC. The use of PMA, able to inhibit amplification of DNA of dead cells, avoids an overestimation of the viable cell number seen with conventional qPCR. However, there are some limitations: the number of cells surviving after treatment should be higher than $10^5$ cells/ml and the treatment should compromise the integrity of the membrane. Nevertheless, the PMA-qPCR method was successfully used to determine the difference in susceptibility of *P. aeruginosa* in a mono- and multispecies biofilm towards colistin and levofloxacin: *P. aeruginosa* grown in a multispecies biofilm appears to be less affected by levofloxacin, and more sensitive to colistin than when grown in a monospecies biofilm. These data indicate that the effect of the presence of different members in a biofilm on the susceptibility of *P. aeruginosa* depends on the antibiotic used, and that *P. aeruginosa* in a multispecies biofilm is not always less susceptible to antibiotics than in a monospecies biofilm.

### Funding

This research has been funded by the Interuniversity Attraction Poles Programme initiated by the Belgian Science Policy Office (P7/28) and by FWO Vlaanderen (3G049412). The funders had no role in study design, data collection and analysis, decision to publish, or preparation of the manuscript.

### Grant Disclosures

The following grant information was disclosed by the authors:
Belgian Science Policy Office: P7/28.
FWO Vlaanderen: 3G049412.

### Competing Interests

Tom Coenye is an Academic Editor for PeerJ.

### Author Contributions

- Sarah Tavernier conceived and designed the experiments, performed the experiments, analyzed the data, contributed reagents/materials/analysis tools, wrote the paper, prepared figures and/or tables.
- Tom Coenye conceived and designed the experiments, contributed reagents/materials/analysis tools, wrote the paper, reviewed drafts of the paper.

### Supplemental Information

Supplemental information for this article can be found online at http://dx.doi.org/10.7717/peerj.787#supplemental-information.

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
