# Peer review of "Quantification of Pseudomonas aeruginosa in multispecies biofilms using PMA-qPCR"

_PeerJ, doi:10.7717/peerj.787_

## Round 0.1 · original submission · Major Revisions

Dear Dr. Coenye,

thank you for submitting your works to PeerJ. We have received overall positive reviews from two reviewers, including some constructive criticism.

The two reviewers shared the view that the manuscript deals with a very important issue in the field of biofilm research, namely to provide a good quantification tool to allow the determination of exact viable numbers of single species within a multispecies population.

They also noted that the method would be far more useful, if several different viable species in the biofilm, in addition to Pseudomonas, could be quantitated at the same time in the same sample. This would provide a broader applicability of the method and allow a better qualitative and quantitative assessment of the biofilm.
The work will also benefit greatly from including spiked or non-spiked test samples of more native materials such as bronchoalveolar lavage or saliva from asymptomatic donors or patients. These additional controls will also dispel possible concerns, whether the sensitivity of the method is susceptible to PMA quenchers or other inhibitory effects present in the native sample (concerns also raised and discussed in the Taylor et al., 2014 study).

A further common note was that recent publications describing and discussing similar methodology (including differentiating viable bacteria by PMA exclusion) for the characterization of biofilm and bacterial mixtures should be correctly referenced (including Alvarez et al., 2013; Yasunaga et al., 2013; Taylor et al., 2014; Krüger NJ et al., 2014)

Legend to Fig. 1, please clarify in the legend which bacterial mixture exactly was used for the experiments displayed in this figure.

Legend to Fig. 2, the type of statistical method performed for this experiment should be stated in the legend.

Please carefully address the few major comments stated by the reviewers and in this letter above, and all minor questions and reviewers’ comments which will help to clarify some possible misunderstandings in the manuscript. The manuscript formatting should be adapted in the corrected version to PeerJ standards.

Reviewer 1 ·

Basic reporting

This method paper by Tavernier and Coenye describes the use of PMA-qPCR to quantitate P. aeruginosa grown in monospecies and multispecies biofilms and subjected to different antibiotics. The paper is very well written, focussed and succinct.

Although this is in part a matter of personal preference, I would suggest to combine Figures 4-6, which all have the same design and show results for three different antibiotics into one figure. Likewise, Tables 1 and 2 could be combined, making it easier for the reader to compare the data for different compounds.

The authors should update the citation list to include other papers reporting use of PMA-qPCR for the analysis of biofilms, such as Yasunaga et al. 2013 and Taylor et al. 2014.

Experimental design

The study is well designed and competently done. It is, however, extremely limited in its technical approach due to the fact that the authors only quantitated P. aeruginosa, even when grown in a multispecies biofilm. Given that all elements of the method (including its limitation for antibiotics not affecting membrane integrity) were described before, the results are largely expected. It would have strengthened the paper if the author had shown the usefulness of multiple (or multiplex) PMA-qPCR to study the composition of a mixed species biofilm (i.e. quantitated the other three species as well). This could be done without much additional effort, if the DNA samples are still available. As it stands, the "multispecies biofilm" is poorly defined. I believe that the other three species were added, but one cannot know if they actually grew.

Validity of the findings

Fig. 1: There are seven lanes on the gel, but only six lanes are specified in the figure legend. Presumably, the 1% mix is missing in the figure legend? All lanes should be labelled in the figure itself, anyway. It seems odd that lanes 4 and 5 in panel B contain less DNA than the neighboring lanes. Was this a reproducible finding, or a pipetting artefact? This figure should only be shown if this is reproducible, or replaced with a figure showing a reproducible gel. Also, no gel should be shown without a size marker.

Reviewer 2 ·

Basic reporting

No major comments on the basic reporting, but please include the published work of Alvarez et. al., (AMB Express. 2013 Jan 4;3(1):1. doi: 10.1186/2191-0855-3-1.Method to quantify live and dead cells in multi-species oral biofilm by real-time PCR with propidium monoazide) in the introduction / discussion.

Experimental design

The manuscript addresses a very important issue in the field of biofilm research, a good quantification tool to allow the determination of exact viable numbers of one single species within a multispecies population. As method of choice the authors chose the relatively new technique of PMA-qPCR and propose it as an alternative to other described methods, highlighting both the strength as well as the weak points of the technology.
This is a timely and important study, however the obtained data (especially those highlighted in fig 1 and 2 ) and their interpretation are critical and seem to contradict results of other studies and the information of the manufacturer (http://biotium.com/technology/pma-for-viability-pcr/) ; Alvarez, AMB Express. 2013 Jan 4;3(1):1. doi: 10.1186/2191-0855-3-1.Method to quantify live and dead cells in multi-species oral biofilm by real-time PCR with propidium monoazide).
Detailed information: see box "Validity of the the findings"

Validity of the findings

Figure 1A nicely shows that PMA-treatment prevents the purification of DNA of dead cells. With increasing percentage of viable cells the DNA band gets stronger, as expected. However, if genomic DNA is not blocked by PMA (Fig. 1B), DNA purification should yield in more or less similar DNA concentrations, independent of the viability state of the cells, but this is not the case. In contrast, dead cells have more genomic DNA than the living ones. This result needs to be explained – especially as it contradicts findings of previous studies/manufacturer’s information.
Minor aspects: The first band is not explained; a mixture of 75% is not mentioned in the M&M.

Figure 2:
It clearly makes sense that the Cq value of living cells (100%) is lower than that of the dead ones (0%) if PMA-treated bars are considered. If there is no DNA available due to the blockage by PMA in the purification step, amplification should be far slower than in the sample with a large amount of genomic DNA (100% alive). This is also in agreement with Figure 1A. However, if samples are not treated with PMA the amounts of genomic DNA should be more or less equal in all samples, independent of the percentage of dead/live cells (see also manufacturer’s homepage (link above)) and thus, lead to similar Cq values.

Figure 3 (minor aspect):
Alvarez and colleagues are able to receive R2 values of 0,998 and higher while the value here is around 0,97. Is the used species/strain an explanation?

Additional comments

Minor general comments:
Abstract: Since the author emphasize the time-consuming factor of cultivation-based methods, it would be nice to know how much time the introduced PMA-qPCR method will take.
p.4, line 60 ff: examples for pPCR papers; cite Alvarez et al 2013
p.5, line 112: What’s the BSA good for?
p.6, line 120f: Since the antibiotics had been solved in PS (instead of medium) why do you expose the cells for 24h to the antibiotics? Growth of the cells shouldn’t be possible so that antibiotics acting on metabolic active cells should not work independent of incubation time.
p.7, line 175f: the overnight culture was higher than OD 0.1 and diluted to 0.1? In what?
p.8, line 182 / Figure 1: one mixture is missing in the text: 75 %
p.8, line 197: Did you choose CFU counts or SPC for the experiment (data not shown)?
p.10, line 260ff: Can you speculate more about the fact that one antibiotic is better active in a mono-species biofilm and the other in a multi-species biofilm? What are consequences for future treatment regimens? Individual resistance profiling and treatment?
Although P. aeruginosa is still the major pathogen in CF, it would have been interesting to see whether PMA-qPCR would have been a useful tool to determine exact numbers for the other species as well. Would the results of PMA-qPCR of all species in sum lead to similar results than determined with other methods?

---

## Round 0.2 · accepted · Accept

Dear Dr. Coenye,

Thank you for resubmitting a revised version of your manuscript.
Almost all reviewers’ comments have been fully addressed. One point that remains for future discussion (was raised by reviewer 2) is the use of PS instead of growth medium for the biofilm antibiotic treatment. It would be good to know whether the use of a substrate medium changes the outcome of any of the antibiotics on the biofilm. Maybe the word “treatment” throughout the text could be replaced by a word that better matches the experimental administration of antibiotics in this setting (such as “experimental administration of antibiotics to the biofilm”).

A few more minor points are remaining before the manuscript will be finally suitable for publication. Feel free to address these comments which will help PeerJ readers to use this paper as a reference method and increase reproducibility of their own data. These changes are of course at your discretion and I fully entrust you with them since the manuscript is accepted. However, in the interest of the future use and reception of your paper, you might agree to include them, since you have provided most of these explanations in your response letter – so why not include them in the text for clarity:

1) Note of caution for readers to be added on page 6 (methods) for clarity: “Viable but not culturable cells of any species might not be detectable by plate counts, which might lead to differences between plate counts and PMA qPCR”.

2) Note of caution to be added to the discussion for clarity: “In native material of multispecies biofilm, for instance from patients’ material, divergent results might occur, and further testing of spiked samples may be used in critical applications to clarify this point of uncertainty also raised by (Taylor et al., 2014)”.

3) I encourage the authors to add a methods’ explanation (provided by the authors themselves in their rebuttal letter), to the methods’ section for clarity, p. 8 paragraph 2.3.5 (methods); in particular, this clarifying addition concerns the results shown in Fig. 4, which still remains somewhat unclear, and an explanation might partially clarify the unclear outcome for the reader: “Actively proliferating bacterial cells have more DNA than inactive or dead cells, which lead to a lower Cq value when the percentage of living cells is increased, even without the addition of PMA. Moreover, the dead cells in the samples were obtained using heat inactivation (95°, 15 min). This heat treatment can have an influence on the degradation of the DNA, and may subsequently lead to a lower amount of intact DNA and thus a higher Cq value when the percentage of inactivated cells increases over live cells (Takahashi et al. 2004).”

4) Please change the last sentence in figure legend to figure 6: “rod-shaped and coccal bacteria are clearly visible in the biofilm image.” Since Gram-pos and Gram-neg bacteria cannot be distinguished by this imaging method.

The manuscript should now be adapted to PeerJ formatting standards for your final resubmission (e.g. regarding the title formats, headers etc.). One reviewer also provided some track changes (some typo corrections) directly in your revised manuscript file, which will be sent to you from the PeerJ editorial office.

Please incorporate these improvements as you see fit at the production stage. You can ask PeerJ associate editor Dr. Sophie Kusy ([email protected]) for assistance with the uploading.

Kind regards,
Christine Josenhans